# Microstructure Characterization and Corrosion Resistance of Zinc Coating Obtained in a Zn-AlNiBi Galvanizing Bath

**Henryk Kania** [1], **Mariola Saternus** [2,*], **Jan Kudláček** [3] and **Jakub Svoboda** [3]

1   Department of Advanced Materials and Technology, Faculty of Engineering Materials,
    Silesian University of Technology, Krasińskiego 8, 40-019 Katowice, Poland; Henryk.Kania@polsl.pl
2   Department of Metallurgy and Recycling, Faculty of Engineering Materials,
    Silesian University of Technology, Krasińskiego 8, 40-019 Katowice, Poland
3   Department of Manufacturing Technology, Czech Technical University in Prague, Technická 4,
    166-07 Prague, Czech Republic; Jan.Kudlacek@fs.cvut.cz (J.K.); jakub.svoboda1@fs.cvut.cz (J.S.)
*   Correspondence: Mariola.Saternus@polsl.pl; Tel.: +48-32-603-4275

**Abstract:** The article examines the impact of the addition of Al, Ni, and Bi to a zinc bath on the microstructure and corrosion resistance of hot dip galvanizing coatings. The microstructure on the surface and the cross-section of the coatings obtained in the Zn-AlNiBi bath were examined. The corrosion resistance of the coatings was assessed by the standard neutral salt spray test (EN ISO 9227), the sulfur dioxide test in a humid atmosphere (EN ISO 6988), and the electrochemical test. The corrosion resistance of Zn-AlNiBi coatings was compared with the corrosion resistance of coatings attained in the bath of "pure" zinc. The corrosion tests showed higher corrosion wear of the coating obtained in the Zn-AlNiBi bath and a higher value of the corrosion current density for this coating. It was found that the cause of the reduction of the corrosion resistance of the coating, in contrast to the coating obtained in the "pure" zinc bath, may be the presence of bismuth precipitates in the coating, which may form additional corrosion cells.

**Keywords:** hot-dip galvanizing; zinc coatings; corrosion resistance

## 1. Introduction

Today, alloy additions to zinc baths are one of the most important factors determining not only the quality of the zinc coating, but also the economics of the galvanizing process itself. The content of the appropriate alloying additives in the bath, and keeping their concentration within the required limits, enables to effectively reduce the amount of waste in the form of galvanizing ashes and hard zinc, but also limit the excessive coating thickness on reactive steels and to reduce the losses caused by insufficient dripping of liquid zinc from the product surface.

The economic benefits resulting from the use of alloying additives for the zinc bath have resulted in the fact that, currently, the industrial galvanizing process is carried out only in multi-component alloy baths. The group of five metals used as alloying additives, i.e., Al, Ni, Pb, Bi, and Sn, has gained dominant importance. Aluminum and nickel are almost always present in the bath, while lead, bismuth, and tin are used interchangeably [1,2].

Aluminum content in the bath from 0.005 wt.%–0.01 wt.% limits the oxidation process of liquid zinc surfaces as a result of the $Al_2O_3$ barrier film formation [3]. In contrast, nickel containing 0.04 wt.%–0.06 wt.% limits the excessive increase of the coating thickness on reactive steels [4,5]. For these metals it is very important to keep their concentration below the upper limit of bath content. Aluminum contents above 0.01 wt.% may cause coating discontinuities as a result of the reaction

of aluminum with flux [6]. Exceeding 0.06 wt.% Ni in the bath increases the amount of hard zinc formed [7].

While the Al and Ni additives are not in doubt, it is more disputable to add Pb, Bi, and Sn to the bath. Lead was present in the bath almost from the beginning of the industrial application of the galvanizing process. Used as an input material for the Good Ordinary Brand (GOB) bath containing up to 1.4% Pb, lead was introduced into the bath in an uncontrolled manner, and its concentration reached a saturation level of 1.2 wt.% [3]. With the use of super high grade (SHG) zinc, lead is a deliberately introduced alloying additive that reduces the surface tension and improves the fluidity of liquid zinc [8,9]. Studies have shown that optimal bath properties are obtained at the content of 0.4 wt.%–0.5 wt.% Pb [10]. However, lead is harmful to the environment and human health [11,12]. Although lead baths are still used, it is increasingly being removed from the bath by replacing it with bismuth. The latter has a number of beneficial properties; it is enough to mention that it causes an almost 10-fold increase in the fluidity of zinc compared to Pb [13], reduces the solubility of iron in zinc [14] and is not harmful to the environment. However, its limitation in the bath is caused by the possible influence of this metal on the occurrence of the liquid metal embrittlement (LME) phenomenon [15]. It is advisable that the entire content of Pb + 10Bi not exceed 1.5 wt.% [16].

Alloying additives to the zinc bath undoubtedly have a positive effect on the quality of coatings and improve the technological process of galvanizing. However, not much research concerns the impact of these additives on the corrosion resistance of coatings. Previous studies have shown an adverse effect of lead on the corrosion resistance of coatings [17–20]. Liu et al. [21], on the other hand, claim that Bi has no effect on the corrosion resistance of coatings. However, some studies indicate that bismuth also has an adverse effect on corrosion resistance [22,23].

This article examines the microstructure of coatings and the corrosion resistance of coatings obtained in a bath containing the optimal content of aluminum, nickel, and bismuth additives, which is currently used as a substitute for lead-containing baths.

## 2. Experimental

### 2.1. Materials and Hot Dipping

The research was aimed at setting down the issue of aluminum, nickel, and bismuth alloying additives for zinc bath on the microstructure and corrosion resistance of the produced coatings. As the input materials for the bath super high grade (SHG) zinc 99.995%, mortars of ZnAl4 and ZnNi0.5 and pure Bi 99.99% were applied. Table 1 presents the chemical composition of the bath, which was assessed with the ARL 3460 emission spectrometer (Thermo ARL, Waltham, MA, USA). The content of the alloying additives in the bath was within the range of the concentration considered to be optimal. In addition, iron was dissolved in the bath until it was saturated with liquid zinc. This enabled to prevent changes in the concentration of iron during tests and to adjust the composition of the bath to real industrial conditions, in which the zinc bath always contains iron. A "pure" zinc bath without alloying additives, also containing iron at the saturation level, was used as the reference bath. The comparative bath tests were performed in real-time and detailed in [17].

**Table 1.** The chemical composition of the zinc baths.

| Bath | Content [wt.%] | | | | | | |
|---|---|---|---|---|---|---|---|
| | Al | Fe | Ni | Pb | Bi | Sn | Zn and Others |
| Zn-AlNiBi | 0.0053 ± 0.0016 | 0.029 ± 0.010 | 0.054 ± 0.0011 | 0.002 ± 0.0002 | 0.0623 ± 0.0012 | 0.0009 ± 0.0001 | residue |
| "pure"zinc [17] | 0.0002 ± 0.0001 | 0.031 ± 0.002 | 0.00010 ± 0.00006 | 0.0013 ± 0.0005 | 0.0003 ± 0.0001 | 0.0007 ± 0.0001 | residue |

Coatings for research were obtained on S235JRG2 low-silicon steel samples with a content of 0.021 wt.% Si. Table 2 shows the chemical composition of the S235JRG2 steel, which was evaluated by a Spectro Lab M8 emission spectrometer (SPECTRO Analytical Instruments, Kleve, Germany). Steel samples measuring 50 mm × 100 mm × 2 mm were immersed for 180 s in a Zn-AlNiBi bath at 450 °C. Before the galvanizing process, the samples were degreased in HydronetDase solution for 5 min, pickled in 12% HCl solution for 10 min, fluxed in TegoFlux60 solution for 2 min and finally dried for 15 min at 120 °C. After the samples were removed from the bath, they were air-cooled. Coatings in the comparative bath of "pure" zinc were made using identical research parameters [17].

**Table 2.** The chemical composition of G235JRG2 steel.

| Grade | Content [wt.%] | | | | | |
|---|---|---|---|---|---|---|
| | C | Si | Mn | S | P | Fe and Others |
| G235JRG2 | 0.13860 ± 0.00607 | 0.02101 ± 0.00070 | 0.74366 ± 0.00278 | 0.00869 ± 0.00034 | 0.00881 ± 0.00043 | residue |

### 2.2. Characterization Methods

Prior to corrosion tests, a preliminary assessment of the quality of the coatings was made based on the disclosure of the structure on the coating cross-section and determination of its average thickness. Metallographic tests were carried out using the Olympus type GX51 light microscope (Tokyo, Japan). AnalySIS software (Olysia m3) was used to record the image. The thickness of the coatings was assessed using an Elcometer 355 magnetic induction meter (Manchester, England). The final result was an average of ten measurements (on each side of the sample).

The microstructure and chemical composition of the coatings were examined applying a Hitachi S-3400 N scanning microscope (Tokyo, Japan) equipped with an energy dispersion spectroscopy X-ray spectrometer with accelerating voltage of 15 kV. Quantitative tests of the chemical composition were carried out on the surface and on the cross-section of the coating obtained in the Zn-AlNiBi bath using the software from Noram Instruments—System Six.

### 2.3. Corrosion Testing Method

#### 2.3.1. Neutral Salt Spray

The tests of resistance to neutral salt spray were carried out according to the ISO 9227 [24] standard in a CORROTHERM Model 610 salt chamber, Erichsen (Hemer, Germany) with a capacity of 400 dm$^3$. A mist of 5% NaCl solution in distilled water was sprayed in the salt chamber, the condensation rate of which on a horizontal surface of 80 cm$^2$ was 1.5 ± 0.5 mL/h. The temperature in the chamber was kept at 35 ± 2 °C, and the pH of the sprayed solution was 6.8–7.2.

Evaluation of the appearance of the samples during the test was carried out every 24 h, while the mass of the samples was measured after 24, 48, 96, 240, 480, 720 and 1000 h of exposure in the chamber. Before measuring the weight, the samples were dried in an oven for 2 h at 30 °C with free air flow to remove moisture from the surface of the samples. The result of the mass measurement was the average of three measurements for five samples of the same type. No corrosion products were removed from the surface of the samples.

#### 2.3.2. Sulfur Dioxide Test in a Humid Atmosphere

Corrosion tests in a humid atmosphere containing SO$_2$ were carried out in keeping with PN EN SO 6988 [25] in a Koesternich Hygrotherm model 519 chamber from Erichsen (Hemer, Germany) with a capacity of 300 dm$^3$. 2 dm$^3$ of distilled water was poured into the chamber for one test cycle and 0.2 dm$^3$ of SO$_2$ was dosed. The research cycle consisted of 8 h exposure of the samples in a

closed chamber and 16 h exposure of the samples in the ambient atmosphere. The temperature was maintained at 40 ± 2 °C during the exposure. After 24 h, the surface condition of the samples was assessed and the mass was measured without removing corrosion products. The result of the mass measurement was the average of three measurements for five samples of the same type.

### 2.3.3. Electrochemical Test

Potentiodynamic tests were carried out to evaluate the electrochemical parameters of the coating corrosion process. The tests were conducted both for the coatings obtained in the Zn-AlNiBi bath, and for the coatings obtained in the "pure" zinc bath. Electrochemical tests were carried out on a PG201 potentiostat/galvanostat from Radiometer (Copenhagen, Denmark). Voltamaster 1 software was used to record the potentiodynamic curves. Potential values were recorded relative to a normal calomel electrode, which was then converted to a normal hydrogen electrode (NHE) shifting the measured values by 244 mV. The active surface of the sample during the test was limited to 1.0 cm$^2$. Before testing, the surface of the samples was degreased with trichlorethylene in an ultrasonic cleaner for 180 s, rinsed in distilled water, and dried in a stream of air. Potentiodynamic tests were carried out at ambient temperature, whereas a 3.5% solution of NaCl in distilled water was used as the electrolyte.

The open cell potential measurement hold out close to 5 min. After evaluating the electrode potential in the studied environment, the reliance of current on potential was registered. Corrosion potential ($E_{corr}$) and corrosion current density ($j_{corr}$) were determined based on the recorded curves by extrapolation of the Tafel line.

## 3. Results and Discussion

### 3.1. Appearance, Structure, and Thickness of Coating

The structure and corrosion resistance of coatings obtained in the Zn-AlNiBi bath were assessed in relation to the coating obtained in the "pure" zinc bath [17]. The adhering coatings had a light and shiny smoothness, which ensured the presence on the surface of the outer layer of the alloy of the zinc bath. The appearance of the surface of the coating obtained in the Zn-AlNiBi bath (Figure 1a) is uniform. The coating shows no lumps or discontinuities. A large spangle is visible on the surface of the coating. On the surface of the coating obtained in the bath of "pure" zinc, especially in the lower part (Figure 1b), a fine crystalline structure is visible. This may indicate insufficient zinc flow from the sample surface and accumulates in the lower part.

Coatings for corrosion tests did not show substantial differences in construction. On the cross-section of the coating obtained in the Zn-AlNiBi bath (Figure 2a), a transition layer constructed of intermetallic phases of the Fe-Zn system can be found, which is covered by the outer layer of the iron solution in zinc η [26]. The outer layer of the coating is actually an alloy layer of the galvanizing bath. In the transition layer, the phase zone $\delta_1$ with a compact structure and uniform thickness is seeable, as well as the ζ phase zone having an uneven separation border with the outer η layer. The comparative coating obtained in the "pure" zinc bath (Figure 2b) has a similar phase morphology on the cross-section, although larger unevenness of the thickness of the ζ phase layer [17] can be observed. Both the structure of the coating obtained in the tested Zn-AlNiBi bath and the comparative coating are characteristic for the coatings obtained on steel with low-silicon.

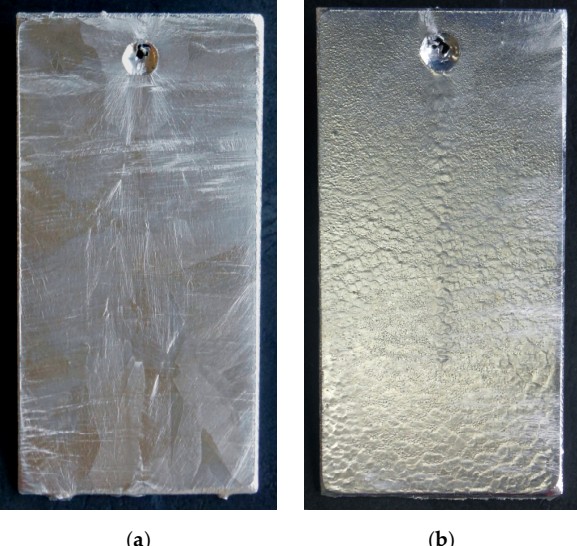

**Figure 1.** The view of the surface of coatings produced in the Zn-AlNiBi (**a**) and "pure" zinc bath (**b**) [17] before the research.

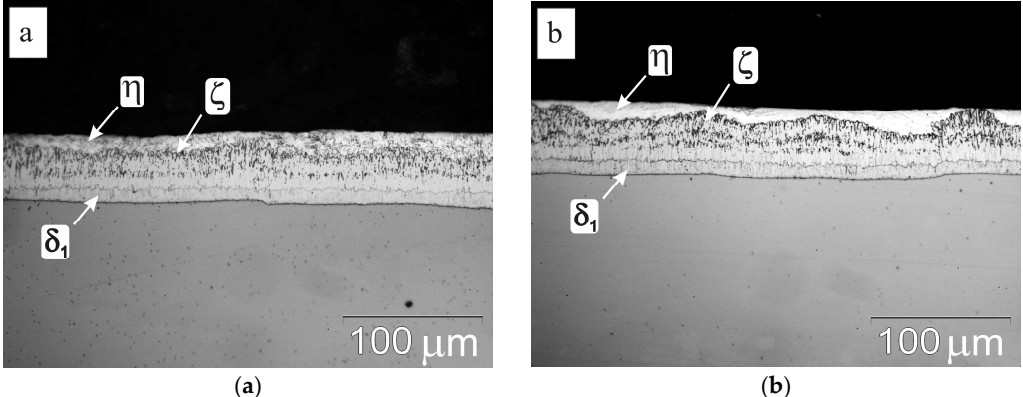

**Figure 2.** The view of a cross-section of coatings produced in Zn-AlNiBi bath (**a**) and "pure" zinc bath (**b**) [17] on samples for corrosion research.

The thickness of the tested coating and comparative coating before corrosion tests are shown in Figure 3. The determined average thickness of the coating obtained in the Zn-AlNiBi bath was 51.09 ± 4.91 μm. A slightly larger thickness of 52.46 ± 6.32 μm was found in the comparative coating obtained in a pure zinc bath [17]. The obtained coatings meet the requirements of EN ISO 1461 [27] in terms of the acceptable and minimum coating thickness. The Ni alloying additive reduces the thickness of the intermetallic phases in the transition layer of the coating [4,5], while Bi is conducive to reducing the thickness of the outer layer of the coating [13]. However, the decrease in coating thickness is relatively small. Based on the assessment of the structure and thickness of the coating, it can therefore be concluded that the studied coating produced in the Zn-AlNiBi bath is very similar in this respect to the comparative coating obtained in the "pure" zinc bath.

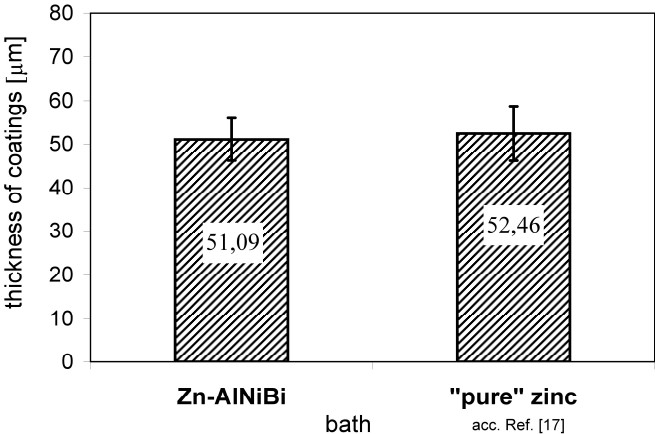

**Figure 3.** The coating thickness produced in Zn-AlNiBi bath and "pure" zinc bath [17] before corrosion research.

*3.2. Surface Microstructure*

The condition of the surface coating has a decisive influence on the corrosion process at its initial stage. Figure 4 shows microstructure of the coating surface produced in the Zn-AlNiBi bath. Table 3 presents the chemical composition in the marked micro-areas. Two morphologically different areas can be seen on the surface of the coating at high magnification. Figure 4a shows areas with a relatively smooth surface (in the middle zone) and areas characterized by banded, parallel irregularities (in the outer zones). On the surface of the coating, the presence of white precipitates of various sizes, unevenly distributed in the matrix can be also noticed. These precipitates have a regular shape. Figure 4b indicates the precipitation with a diameter of 4.1 μm. The microanalysis of the chemical composition carried out in the precipitation marked in Figure 4c as point 2 showed the presence of 76.7 wt.% Bi and 23.3 wt.% Zn (Table 3). Zinc at 100 wt.% was found in the matrix (point 1, Table 3). The coating obtained in the "pure" zinc bath (Figure 4c) shows a similar, heterogeneous surface morphology, and its chemical composition was found to be 100 wt.% zinc (point 3, Table 3). Therefore, the initial state of the surface of the tested Zn-AlNiBi coating differs from the state of the surface of the comparative coating obtained in the "pure" zinc bath by the presence on the surface of Bi precipitates. In accordance with the Bi-Zn equilibrium system, both Bi in Zn and Zn in Bi do not show solubility in the solid state [28,29]. It can therefore be taken for granted that bismuth is separated from the solid zinc solution, forming precipitates of this metal in the zinc matrix.

**Table 3.** The chemical composition in chosen micro-areas on the surface of the Zn–AlNiBi coating (analysis points marked on Figure 4).

| Analyzing Points | Element Contents/Error (+/−1 Sigma) | | | | | | | |
| | Zn-K | | | | Bi-M | | | |
| | wt.% | wt.% Error | at.% | at.% Error | wt.% | wt.% Error | at.% | at.% Error |
|---|---|---|---|---|---|---|---|---|
| point 1 | 100 | ±1.8 | 100 | ±1.8 | – | – | – | – |
| point 2 | 23.3 | ±1.4 | 49.3 | ±2.7 | 76.7 | ±0.7 | 50.7 | ±0.5 |
| point 3 | 100 | ±1.6 | 100 | ±1.6 | – | – | – | – |

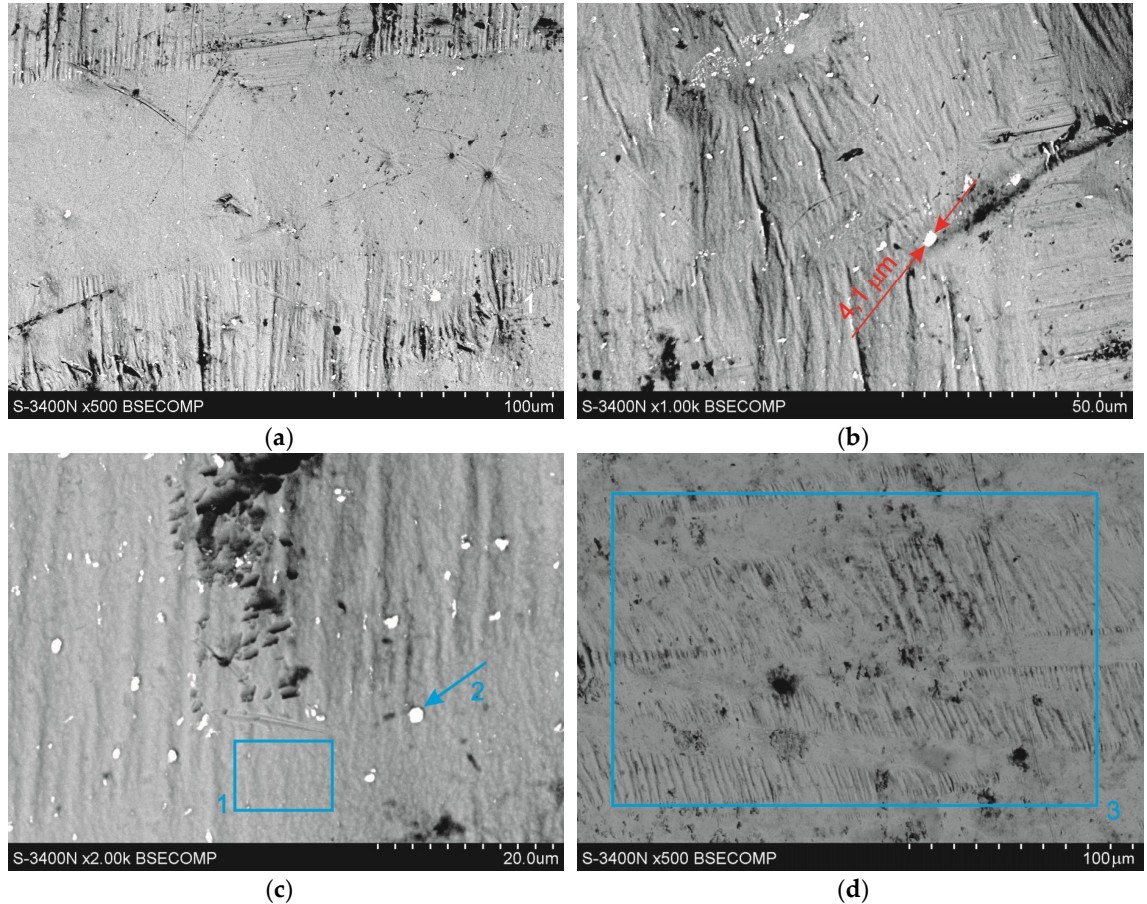

**Figure 4.** Microstructure (SEM) of the Zn-AlNiBi coating surface: (**a**) sight of the coating surface, (**b**,**c**) bismuth precipitation on the top surface, and the microstructure (SEM) of the "pure" zinc coating surface (**d**).

### 3.3. Cross-Sectional Microstructure of Coating

Figure 5 presents the microstructure on the Zn-AlNiBi coating cross-section, while the chemical composition in selected micro-areas of the coating is summarized in Table 4. The coating has a typical layered construction, characteristic of low-silicon steel. In the outer layer η, 100 wt.% Zn was found (Figure 5a, point 4, Table 4). The chemical composition of the structural components of the diffusion layer is consistent with the content of Fe and Zn in the intermetallic phases of the Fe-Zn system [30]. The ζ phase layer has two distinct zones. In the outer zone of heterogeneous structure, ζ phase has the form of loosely packed crystals, in which the content of 5.4 wt.% Fe and 94.6 wt.% Zn (Figure 5b, point 7, Table 4) has been determined. The internal zone of the ζ phase with a compact structure contains 6.1 wt.% Fe and 93.9 wt.% Zn (Figure 5b, point 8, Table 4). The presence of two zones in the ζ phase layer results from the mechanism of its formation. The layer of this phase increases in contact with liquid zinc as a result of diffusive growth, leading to the formation of a compact zone and the dissolution of this zone and secondary crystallization from an iron-saturated zinc solution, which leads to the formation of a heterogeneous zone [31]. The chemical composition specified at point 9 (9.3 wt.% Fe and 90.7 wt.% Zn) is consistent with the content of ingredients in the $\delta_1$ phase, while at point 10 (22.43 wt.% Fe and 77.6 wt.% Zn) it corresponds to the content of ingredients in the phase Γ (Figure 5b, Table 4).

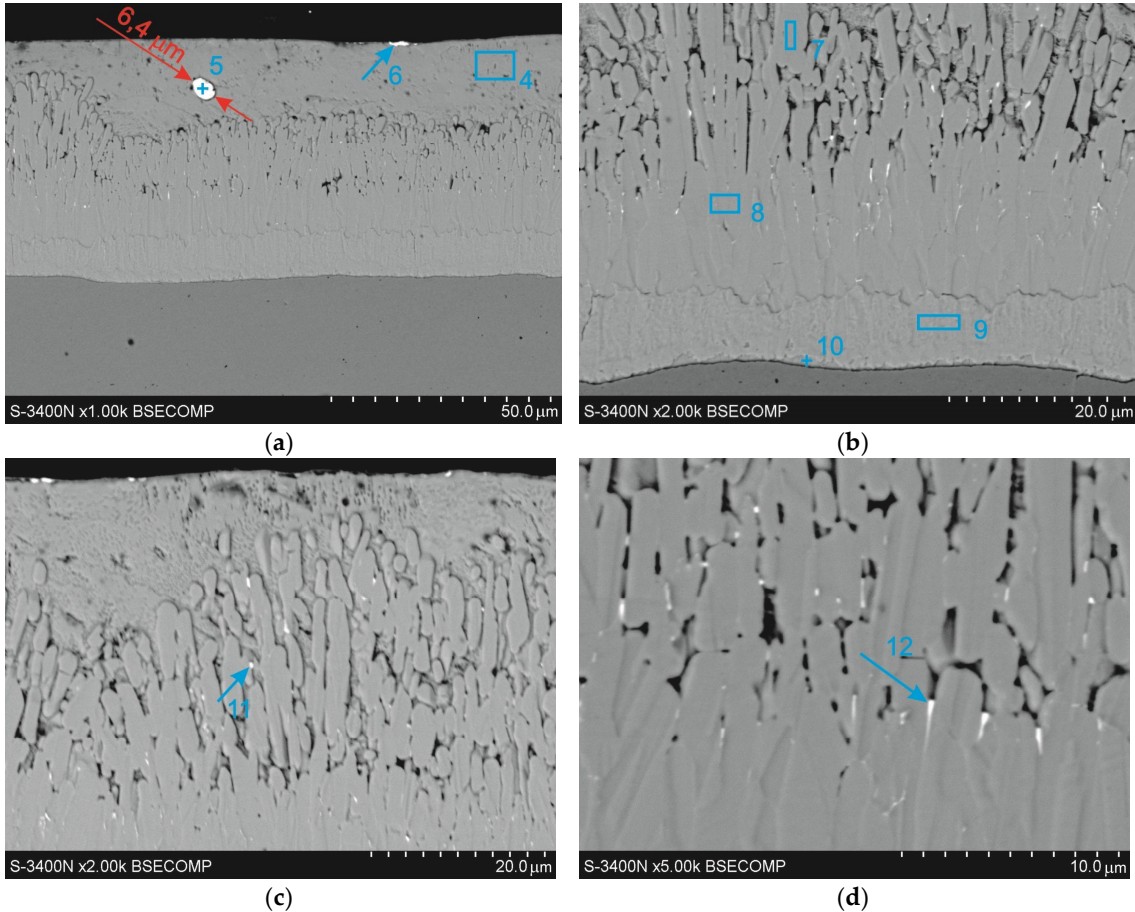

**Figure 5.** Microstructure (SEM) of the Zn–AlNiBi coating cross-section: (**a**) sight of the cross-section of the coating with large precipitation in the outer layer; (**b**) view of the diffusion layer of the coating; (**c**) bismuth precipitation on ζ phase crystals in a zone of heterogeneous structure; (**d**) bismuth precipitation in the ζ phase of the compact structure.

The chemical composition of the identified phases did not reveal the presence of Al and Ni contained in the galvanizing bath or their concentration is below the measuring accuracy of the EDS method. However, microanalysis of the chemical composition made it possible to find Bi in the form of precipitates also on the cross-section of the coating. Bismuth creates precipitations in the outer η layer, but also in the diffusion layer of the coating. In the outer η layer there are individual precipitates of regular shape and relatively large sizes. Figure 5a shows a precipitation with a diameter of 6.4 μm, whose chemical composition (87.2 wt.% Bi and 12.8 wt.% Zn in point 5) allows to state that it is the precipitation of Bi in the η phase matrix. Precipitations on the surface of the coating have a clearly flat shape (Figure 5a, point 6).

In the diffusion layer, the Bi precipitations are much smaller in size and their distribution is more even (fine white precipitations in Figure 5c,d). The preferred place for crystallization of the precipitates in the zone with heterogeneous structure of the ζ phase layer are the crystal boundaries. At point 11, the occurrence of 11.7 wt.% Bi, 3.6 wt.% Fe and 84.7 wt.% Zn (Figure 5c, Table 4) was found, which with a relatively small size of the precipitation allows to state that it is a Bi precipitation in the area of the intermetallic phase of the Fe-Zn system. This zone has a relatively loose structure, and the spaces between the ζ phase crystals can be filled with the η phase, in which Bi can easily be released in the solid state. The concentration of precipitates in the zone of the ζ phase layer with a compact structure drops significantly. Precipitates of bismuth appear rather at the border with the zone of heterogeneous structure and are tightly packed between the ζ phase crystals (Figure 5d, point 12, Table 5—15.1 wt.% Bi, 4.4 wt.% Fe, and 80.5 wt.% Zn). The compact structure of the ζ phase zone makes it very difficult

to locate Bi precipitates in it. No bismuth precipitates were found in the $\delta_1$ phase layer, which also has a compact structure. The location of the precipitates and the lack of Bi content in the chemical composition of the $\zeta$ and $\delta_1$ phases may indicate that Bi is not soluble in the phases of the Fe-Zn system. At present, however, there is a lack of data in the literature confirming both the solubility of Bi as well as its lack in the intermetallic phases of the Fe-Zn system. Considering that Bi exhibits unlimited solubility in liquid zinc [28,29] and assuming its absence in the intermetallic phases of the Fe-Zn system, it can be assumed that the release of Bi during the crystallization of the $\eta$ phase is more likely than its release during growth of $\zeta$ and $\delta_1$ phases. However, the consumption of zinc for the increase of $\zeta$ and $\delta_1$ phases may lead to an increase in Bi concentration in the $\eta$ phase zone.

**Table 4.** The chemical composition in chosen micro-areas on the Zn-AlNiBi coating cross-section (analysis points marked on Figure 5).

| Analyzing Points | Element Contents/Error (+/− 1 Sigma) | | | | | | | | | | | |
|---|---|---|---|---|---|---|---|---|---|---|---|---|
| | Zn-K | | | | Fe-K | | | | Bi-M | | | |
| | wt.% | wt.% Error | at.% | at.% Error | wt.% | wt.% Error | at.% | at.% Error | wt.% | wt.% Error | at.% | at.% Error |
| point 4 | 100 | ±2.4 | 100 | ±2.4 | – | – | – | – | – | – | – | – |
| point 5 | 12.8 | ±1.2 | 31.9 | ±1.6 | – | – | – | – | 87.2 | ±0.8 | 68.1 | ±0.6 |
| point 6 | 37.7 | ±1.2 | 65.9 | ±2.2 | – | – | – | – | 62.3 | ±0.5 | 34.1 | ±0.3 |
| point 7 | 94.6 | ±2.1 | 93.7 | ±2.1 | 5.4 | ±0.4 | 6.3 | ±0.5 | – | – | – | – |
| point 8 | 93.9 | ±2.0 | 92.9 | ±1.9 | 6.1 | ±0.2 | 7.1 | ±0.3 | – | – | – | – |
| point 9 | 90.7 | ±2.0 | 89.3 | ±2.0 | 9.3 | ±0.5 | 10.7 | ±0.5 | – | – | – | – |
| point 10 | 77.6 | ±1.6 | 74.7 | ±1.5 | 22.4 | ±0.5 | 25.3 | ±0.6 | – | – | – | – |
| point 11 | 84.7 | ±1.9 | 91.5 | ±2.0 | 3.6 | ±0.4 | 4.6 | ±0.5 | 11.7 | ±0.4 | 4.0 | ±0.1 |
| point 12 | 80.5 | ±1.6 | 89.1 | ±1.9 | 4.4 | ±0.4 | 5.7 | ±0.5 | 15.1 | ±0.4 | 5.2 | ±0.2 |

**Table 5.** Corrosion parameter values determined from polarization curves.

| Coating | $j_{corr}$ (mA/cm$^2$) | $E_{corr}$ (mV vs. NHE) |
|---|---|---|
| Zn-AlNiBi | −14.26 | −779.67 |
| "pure" zinc | −10.18 | −766.26 |

*3.4. Effects of Corrosion Resistance Tests*

The corrosion resistance of coatings obtained in the Zn-AlNiBi bath was compared with the corrosion resistance of the coating obtained in the "pure" zinc bath [17], the exposure of which in the corrosion test in the neutral salt spray and sulfur dioxide test in a humid atmosphere took place at the same time. The electrochemical corrosion parameters were determined on the basis of potentiodynamic tests in a 5% NaCl solution.

3.4.1. Corrosion Resistance Determined via the Neutral Salt Spray Test

In the salt chamber, the degree of coating wear was assessed on the basis of observations of changes in the coating surface and the measurements of mass changes during the corrosion test. The outlook of the coatings' surfaces after the corrosion test (1000 h of exposure in the salt chamber) is shown in Figure 6. The coating obtained in the Zn-AlNiBi bath was covered mainly with white corrosion products, while red corrosion products were also seen on its surface (Figure 6a). The upper part of the sample was corroded particularly intensively, showing single, pierced points of the coating to the substrate (marked in yellow) (Figure 6a). Similar corrosion products are visible on the comparative coating obtained in the bath of "pure" zinc (Figure 6b). In this case, it can be seen that the proportion of red corrosion products is much smaller and they are more evenly distributed over the surface of the coating. The formation of red corrosion products is caused by the ongoing iron corrosion processes [32]. White corrosion products result from zinc corrosion [33]. Corrosion of the intermetallic phases of the

Fe-Zn system is accompanied by the formation of white corrosion products and red corrosion products of relatively low color intensity [34]. The surface appearance after 1000 h exposure in a salt chamber, therefore, suggested that corrosion occurs in the diffusion layer of the coating. The coating obtained in the Zn-AlNiBi bath showed a very similar cross-sectional structure (Figure 2) and a similar thickness (Figure 3) to the comparative coating obtained in the bath of "pure" zinc. The greater share of red corrosion products on the surface, and the presence of local, puncture penetrations to the substrate, indicates a more intense course of corrosion of the coating obtained in the bath containing Al, Ni, and Bi additives.

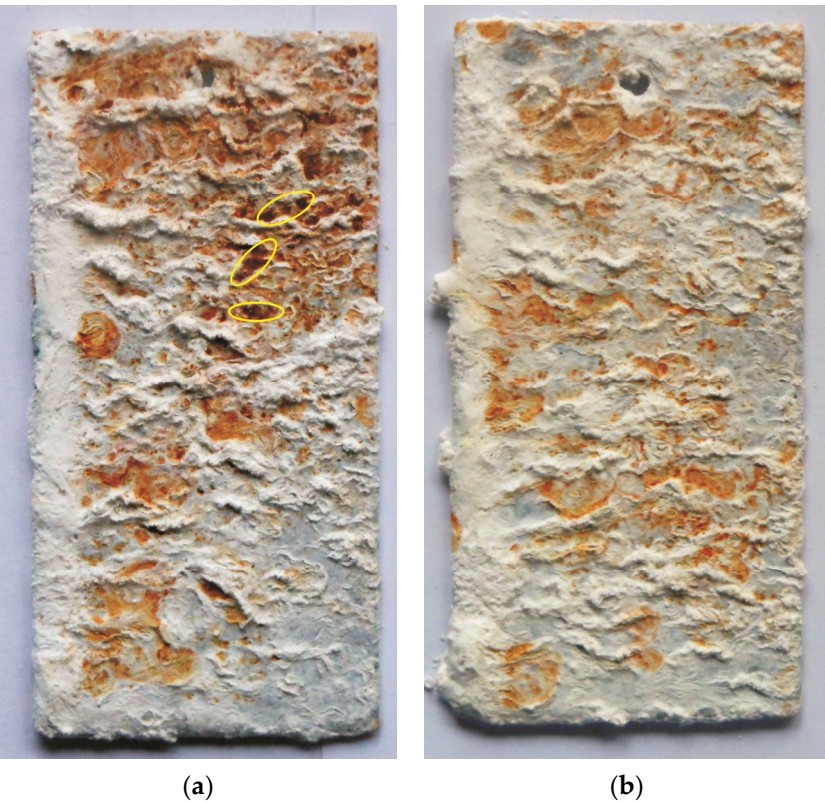

(**a**)                    (**b**)

**Figure 6.** The view of the surface of: Zn-AlNiBi coating (**a**) and "pure" zinc coating (**b**) [17] after 1000 h of exhibition in a salt chamber.

The coating obtained in the ZnAlNiBi bath during the corrosion test in the salt chamber showed a weight gain. Figure 7 shows a comparison of the unitary mass increase of the tested coating with the unitary mass increase of the mass obtained in the bath of "pure" zinc. The coating obtained in the Zn-AlNiBi bath showed greater weight gain from the beginning of the corrosion test. After the tests in the salt chamber (after 1000 h), the coating showed a unitary mass change of 140.34 $g/m^2$. At that time, the unitary mass change of the coating obtained in the "pure zinc" bath was 108.24 $g/m^2$. The amount of corrosion products of the coating obtained in the Zn-AlNiBi bath increased by almost 30%, which confirms the higher intensity of corrosion compared to the coating obtained in the bath of "pure" zinc.

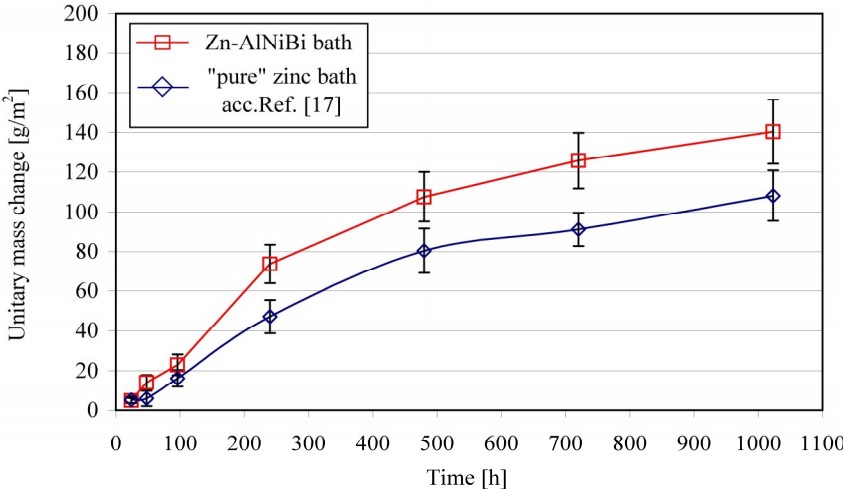

**Figure 7.** Unitary changes in the mass of Zn-AlNiBi and Zn coatings during the test in neutral salt spray.

A parallel conducted corrosion test [17] in neutral salt spray (NSS test) (ISO 9227) [24] of the coating obtained in a Zn-AlNiPb bath (0.0048% Al, 0.049% Ni, 0.48% Pb) [9] showed after 1000 h of exposure a mass change equal to 157.42 g/m$^2$. The tested Zn-AlNiBi bath contained a comparable content of Al (0.0053 wt.%) and Ni (0.054%) and a lower content of Bi (0.062%) than the content of Pb. The smaller increase in the mass of the coating obtained in the bath containing Bi with almost 10-times lower content in the bath does not clearly confirm the lower influence of Bi on the corrosion resistance of coatings compared to Pb. However, taking into consideration the technological aspects of the galvanizing process, and assuming that the baths contained the optimal concentration of Bi and Pb, it could be argued that the Bi content would reduce the corrosion resistance to a lesser extent than the optimal Pb content in the bath.

### 3.4.2. Corrosion Resistance Determined via the Sulfur Dioxide Test in a Humid Atmosphere

The view of the surface of the Zn-AlNiBi coating after the corrosion test in a humid atmosphere containing sulfur dioxide is presented in Figure 8a. It could be noticed that after 30 test cycles the coating revealed no penetration into the substrate. It has a gray and matte appearance, and the large spangle on its surface has become more pronounced. The comparative zinc coating had a similar gray and matte appearance. A fine-grained structure appeared on its surface (Figure 8b). The presence of spangle and visible zinc grains on the coatings surface indicates that the corrosion process took place in the outer layer of the coating.

However, the appearance of the coating surface after the corrosion test does not enable a comparison of the corrosion resistance. The unitary mass changes recorded during the test (Figure 9) indicate, however, greater losses in mass of the coating obtained in the Zn-AlNiBi bath. This coating shows comparable, and even slightly less, mass losses than a coating obtained in a bath of "pure" zinc for up to about 13 cycles. Further extension of the corrosive time accelerates the corrosion process with a tendency to increase the difference in mass loss of the compared coatings. After testing in a humid atmosphere containing SO$_2$, the unitary mass change of the Zn-AlNiBi coating was 24.56 g/m$^2$, while the mass change of the pure Zn coating was 18.04 g/m$^2$. Considering that, in this corrosion test, the corrosion process took place in the outer layer, it can be argued that the outer layer of the coating obtained in the Zn-AlNiBi bath undergoes much more extensive dissolution. For the coating obtained in the alternative Zn-AlNiPb bath in the parallel conducted test [17], the unitary mass change was determined at 21.56 g/m$^2$. In the case of a corrosion test in a humid atmosphere containing SO$_2$, it can therefore be unequivocally stated that the addition of Bi to the zinc bath, even at 10-times lower concentration, significantly accelerates the corrosion of the coating than the addition of Pb.

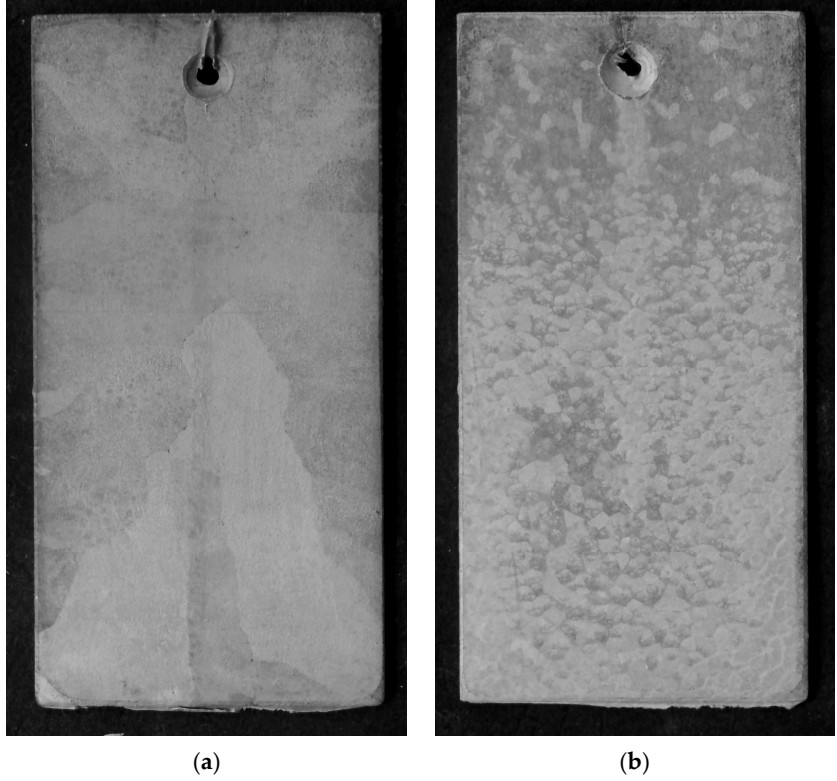

(**a**)                    (**b**)

**Figure 8.** The view of the surface of zinc coatings obtained in: Zn-AlNiBi (**a**) and pure zinc (**b**) [17] baths after 30 test cycles in a sulfur dioxide test in the humid atmosphere.

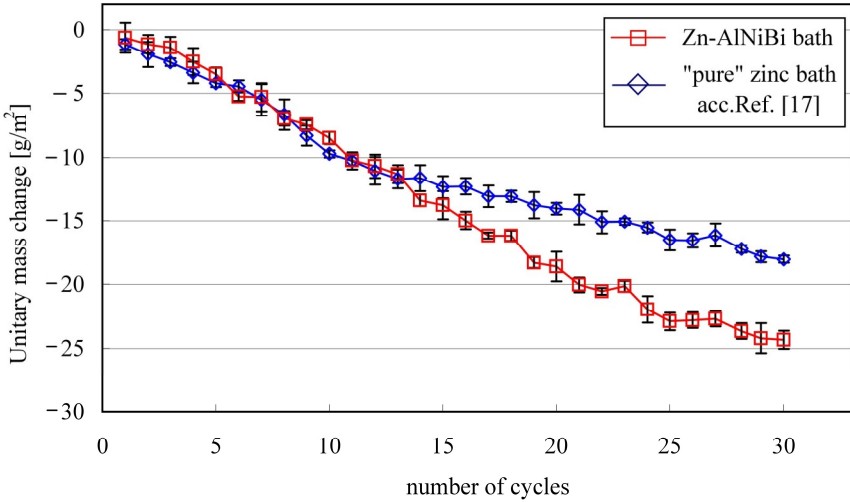

**Figure 9.** Unitary changes in the mass of "pure" zinc and Zn-AlNiBi coatings during the sulfur dioxide test in a humid atmosphere.

### 3.4.3. Corrosion Resistance Determined via an Electrochemical Test

Corrosion current density ($j_{corr}$) and corrosion potential ($E_{corr}$) of the tested coatings were determined by extrapolation of Tafel straight line polarization curves on the anode and cathode side [35]. The course of polarization curves in a 3.5% NaCl solution for coatings made in the Zn-AlNiBi bath and in the "pure" zinc bath are respectively shown in Figure 10. The cathode and anode polarization curves for the coating obtained in the Zn-AlNiBi bath are shifted to higher values of current density. The determined values of corrosion current density ($j_{corr}$) are −14.26 mA/cm$^2$ for the coating obtained in the Zn-AlNiBi bath and −10.18 mA/cm$^2$ for the coating obtained in the "pure

zinc" bath (Table 5), respectively. According to Faraday's law, the corrosion current density ($j_{corr}$) is proportional to the amount of mass changes [36]. The higher value of the corrosion current density for the coating obtained in the Zn-AlNiBi bath indicates its more intense dissolution and lower corrosion resistance in a solution of 3.5% NaCl. The graph also shows a slight shift in the corrosion potential ($E_{corr}$) towards negative values. A lower potential may indicate a higher corrosion tendency of the coating obtained in the Zn-AlNiBi bath. The corrosion potentials of the coating obtained in the Zn-AlNiBi and Zn bath (Table 5) are −779.67 and −766.26 mV respectively (vs. NHE). In such conditions, the tested coatings provide sacrificial protection for steel [37–41].

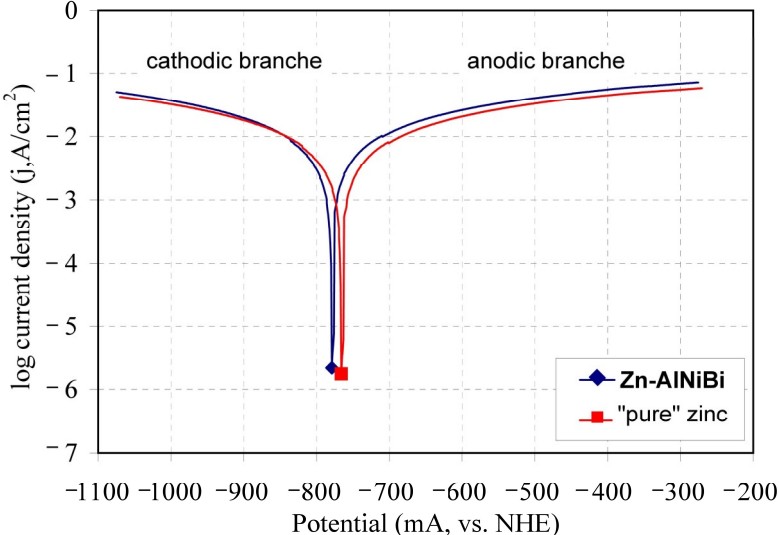

**Figure 10.** Polarization curves of coatings obtained in Zn-AlNiBi and "pure" zinc baths.

## 4. Conclusions

The effect of the addition of Al, Ni, and Bi to the zinc bath on the microstructure and corrosion resistance of coatings have been studied. Based on the conducted research, the following conclusions were formulated:

(1) In the microstructure of the coating obtained in the Zn-AlNiBi bath, the presence of large Bi precipitates in the outer layer on its surface and cross-section and small Bi precipitates in the ζ phase layer was found. No other bath alloying additives (Al and Ni) were found in the coating.

(2) Corrosion tests carried out with a neutral salt spray, in a humid atmosphere containing sulfur dioxide, and potentiodynamic tests consistently confirm the reduction of corrosion resistance of coatings obtained in the Zn-AlNiBi bath in comparison with the corrosion resistance of coatings obtained in the bath of "pure" zinc.

(3) The reason for the decrease in corrosion resistance may be the presence of Bi precipitates in the coating, which, having a positive standard potential, creates a corrosive cell with Zn, accelerating the corrosion process of the coating.

In summary, it can be stated that the Bi bath additive, currently used instead of the environmentally harmful Pb additive, reduces the corrosion resistance of the coating.

**Author Contributions:** Conceptualization, H.K.; methodology, H.K. and M.S.; validation, J.K.; investigation, H.K. and M.S.; resources, J.S.; writing—original draft preparation, H.K.; writing—review and editing, M.S.; visualization, M.S.; supervision, J.K. All authors have read and agreed to the published version of the manuscript.

**Funding:** This research received no external funding.

**Acknowledgments:** The research was supported with students by project SGS19/163/OHK2/3T/12 Research, Optimization, and Innovation of Production Processes.

**Conflicts of Interest:** The authors declare no conflict of interest.

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
