# Peer review of "Microstructure Characterization and Corrosion Resistance of Zinc Coating Obtained in a Zn-AlNiBi Galvanizing Bath"

_coatings, doi:10.3390/coatings10080758_

Round 1
Reviewer 1 Report
In this work, the authors have done much work to study the coating as well as its corrosion resistance. However, the good resistance of coating is the key. The experimental results showed that the addition of AlNiBi did not improve the corrosion resistance of the coating. So, it is not clearly what is the Synergistic effect of AlNiBi addition. Some other comments are as below:
1. In Fig. 2, the EDS mapping of SEM images should be given to verify the composition of the different coating.
2. From Fig. 4, it is seen that the surface of all coating has amounts of defects, and the defects can be the key reasons leading to the lower corrosion protective effects of coating. So, the authors should focus on the integrity of the surface film.
3. In Fig.7, the unitary mass change is not clear. Does the mass contain water?
4. The authors only give the polarization curves. However, the change of OCP and EIS are commonly used to evaluate the coating.
5. The discussion in this work is poor.
Author Response
Dear Reviewer,
We are grateful for taking your time to read our paper and for their constructive comments. We have carefully reviewed the comments and have revised the manuscript accordingly. Our responses are below given in a point-by-point manner. Changes to the text are shown in red in the revised manuscript. We hope the revised version is now suitable for publication.
In this work, the authors have done much work to study the coating as well as its corrosion resistance. However, the good resistance of coating is the key. The experimental results showed that the addition of AlNiBi did not improve the corrosion resistance of the coating. So, it is not clearly what is the Synergistic effect of AlNiBi addition.
Increasing corrosion resistance is the main purpose of using zinc coatings. The studied configuration of AlNiBi alloying additions is relatively often used in industrial practice mainly to improve the galvanizing process technology and reduce the reactivity of steel in liquid zinc. Despite the benefits of using these alloying additions in research, their influence on the corrosion resistance of coatings is ignored and their impact in this respect is not well recognized. Therefore, the research presented in the article was undertaken. As the Al, Ni and Bi additions are not used individually, their synergistic effect on corrosion resistance was therefore investigated. The tests did not show the presence of Al and Ni in the coating, so the key alloying additive affecting the corrosion resistance turned out to be Bi, which negatively affects it. However, the presence of Al and Ni in zinc baths cannot be omitted in the tested alloy additive system.
However, we believe that the reviewer's remark about the lack of clarity of the synergy effect is correct. Taking into account the obtained research result, the word "Synergistic" used in the title of the article is inappropriate as it does not fully characterize the content of the article.
Therefore, we changed the title of the article:
“Microstructure characterization and corrosion resistance of zinc coating obtained in
Zn-AlNiBi galvanizing bath”.
Some other comments are as below:
1. In Fig. 2, the EDS mapping of SEM images should be given to verify the composition of the different coating.
Fig. 2 shows the structure of the coatings revealed on a light microscope. The disclosure of the structure was to compare the structure of coatings before corrosion tests. The morphology of the disclosed phases in the coatings is typical and has been described referring to the literature. Light microscopy does not provide the opportunity to reveal all the details of the structure, e.g. Bi precipitates, and to perform EDS chemical composition studies. Therefore, structural tests (SEM) were performed at high magnifications, at the same time specifying EDS, which are presented for the tested coating in Fig. 5 and in Table 4.
2. From Fig. 4, it is seen that the surface of all coating has amounts of defects, and the defects can be the key reasons leading to the lower corrosion protective effects of coating. So, the authors should focus on the integrity of the surface film.
The condition of the surface is very important for the corrosion process, especially at its initial stage.
The surface of the coating presented in Fig. 4a, b, c is a typical condition of the surface of the coating obtained under the conditions of the hot dip galvanizing process. Indeed, according to the reviewer's comment, the presented results give the impression that this condition may affect the course of corrosion. Therefore, the test results were supplemented by also showing the state of the surface of the comparative coating obtained in a pure zinc bath (Fig. 4d), which shows similar defects on the surface. A comment was also added on lines 192-196.
3. In Fig.7, the unitary mass change is not clear. Does the mass contain water?
According to the reviewer's comment, corrosion products may contain water immediately after the salt spray test. The measurement was carried out on the weight with an accuracy of 0.00001 [g]. Measurement on the weight with such accuracy would be unstable if the samples contained moisture. That is why the samples after being removed from the salt chamber were thoroughly dried. The research methodology has supplemented the text with this information in lines 116-117.
4. The authors only give the polarization curves. However, the change of OCP and EIS are commonly used to evaluate the coating.
We share the reviewer's remark regarding OCP and EIS research. However, the planned experiment was to verify the corrosion resistance of zinc coatings in standard corrosion tests that are used to test zinc coatings as acceptance tests. Tests in the salt chamber and in the Koesternich chamber allow for simulation of long-term corrosion properties of coatings, which cannot be provided by electrochemical tests. Therefore, polarization curve tests only complement the other corrosive tests. In the authors' opinion, this scope of tests allows for a relatively effective assessment of the resistance of the tested coatings.
5. The discussion in this work is poor.
The paper refers to the currently available literature on the impact of alloying additives on corrosion resistance. In the literature, most of the data presents the benefits of alloying additions related to shaping the structure of coatings and the process technology of galvanizing. Little research concerns the impact of these additives on corrosion resistance. In addition, there was a discussion comparing the test bath with an alternative Zn-AlNiPb bath. The bath containing Pb is now increasingly being replaced with a bath containing Bi due to Pb toxicity.
Reviewer 2 Report
The authors present the effect of Bi, Ni, and Al addition to hot-dip Zn bath coatings on the corrosion resistance using a standard neutral salt spray test (EN ISO 9227), sulfur dioxide test in a humid atmosphere (EN ISO 6988) and electrochemical test.
The article is well written, presents new insights in this field, hence I suggest to publish the work after revising following points:
1. Please indicate the experimental errors for all data where applicable (including EDS and the elemental composition measured by the emission spectrometer). I see this as a minimum standard for scientific work, hence I recommend this as a mandatory point before publication.
2. Please replace the “u” by “µ” in your SEM micrographs.
3. The authors are repeatedly comparing the results of Al, Ni, Bi addition with those of Al, Ni, Pb additives of their previous paper (reference [17]). As this work is also published in an MDPI journal a direct comparison by using reprints instead of just citing the article would help the reader to keep track of the findings and increase the readability of this work.
Author Response
Dear Reviewer,
We are grateful for taking your time to read our paper and for their constructive comments. We have carefully reviewed the comments and have revised the manuscript accordingly. Our responses are below given in a point-by-point manner. Changes to the text are shown in red in the revised manuscript. We hope the revised version is now suitable for publication.
The article is well written, presents new insights in this field, hence I suggest to publish the work after revising following points:
1. Please indicate the experimental errors for all data where applicable (including EDS and the elemental composition measured by the emission spectrometer). I see this as a minimum standard for scientific work, hence I recommend this as a mandatory point before publication.
As suggested by the reviewer, measurement errors in tables 1, 2, 3 and 4 have been supplemented.
2. Please replace the “u” by “µ” in your SEM micrographs.
The scale on SEM micrographs and photos from a light microscope is automatically plotted by a computer program. According to the reviewer's comment, the editing error in the pictures (Fig. 4 and Fig. 5) has been corrected. Also Figures 2 a and b were corrected
3. The authors are repeatedly comparing the results of Al, Ni, Bi addition with those of Al, Ni, Pb additives of their previous paper (reference [17]). As this work is also published in an MDPI journal a direct comparison by using reprints instead of just citing the article would help the reader to keep track of the findings and increase the readability of this work.
Comparison of the results of studies on Zn-AlNiBi baths with a Zn-AlNiPb bath is justified, as these baths are currently used alternatively. In comparison, the values from Zn-AlNiPb baths conducted and published earlier were quoted directly. The article about the ZnAlNiPb bath is widely available to the reader, hence the research results were not included in the article. The authors also believe that posting these results in the article would give the impression of presenting the same, already published research results.
Reviewer 3 Report
The authors reported results on the effect of the addition of Al, Ni and Bi to a zinc bath on the microstructure and corrosion resistance of prepared galvanizing coatings. Their Corrosion tests showed a higher corrosion wear of prepared coatings in the Zn-AlNiBi bath compared to those prepared in a bath of pure zinc. The authors explain this result by the bismuth precipitates in the coating, which may form additional corrosion cells.
The authors could take into account the below points for the revised version of the paper:
1. The paper needs language editing.
2. The authors could explain the synergetic effect from their results, as it is not very clear from the manuscript.
3. The authors should discuss their results in the light of the measurements errors when comparing with those of other authors.
The present work may be accepted for publication, after revision and incorporation of the above corrections and comments.
Author Response
Dear Reviewer,
We are grateful for taking your time to read our paper and for their constructive comments. We have carefully reviewed the comments and have revised the manuscript accordingly. Our responses are below given in a point-by-point manner. Changes to the text are shown in red in the revised manuscript. We hope the revised version is now suitable for publication.
1. The paper needs language editing.
The article has been linguistically improved.
2. The authors could explain the synergetic effect from their results, as it is not very clear from the manuscript.
The studied configuration of AlNiBi alloying additives is relatively often used in industrial practice, mainly in order to improve the processability of the galvanizing process and reduce the reactivity of steel in liquid zinc. In addition, Bi in combination with Al and Ni is now more and more often a replacement for Zn-AlNiPb baths containing toxic Pb. As the additives Al, Ni and Bi are not used individually, their combined effect on corrosion resistance was investigated. The tests did not show the presence of Al and Ni in the coating, and the key alloy additive influencing the corrosion resistance turned out to be the addition of Bi, which has a negative effect on it.
In the tested system of alloying elements, the presence of Al and Ni in the bath cannot be ignored. Therefore, the assumption was to investigate the synergistic effect of the AlNiBi additive.
However, we believe that the reviewer's remark about the lack of clarity of the synergy effect is correct. Taking into account the obtained research result, the word "Synergistic" used in the title of the article is inappropriate as it does not fully characterize the content of the article.
Therefore, we changed the title of the article:
“Microstructure characterization and corrosion resistance of zinc coating obtained in
Zn-AlNiBi galvanizing bath”.
3. The authors should discuss their results in the light of the measurements errors when comparing with those of other authors.
The research conducted so far on the influence alloying additives to batch hot dip galvanizing bath presented in the literature concerns mainly the influence of these alloying additives on the growth of intermetallic phases of the Fe-Zn system in the coating, taking into account the reactivity of steel, and their influence on the technological process of galvanizing is described. The problem of the influence of alloy additives to the zinc bath on corrosion resistance is practically ignored in the research. Few publications deal with the effect of these additives on corrosion resistance.
The article refers to the currently available literature on the influence of alloying additives on the corrosion resistance of batch hot hip galvanizing coatings. Therefore, the authors do not understand how to discuss the obtained results of corrosion resistance tests in the light of the measurement errors of other authors, if there are no other alternative test results.
Round 2
Reviewer 3 Report
I am satisfied with the author’s modifications and responses to the reviewer comments, I think that the manuscript in its present form deserve to be published in coatings journal.